# Ultrasound-Derived Skinfolds in Anthropometric Predictive Equations Overestimate Fat Mass: A Validation Study Using a Four-Component Model

**DOI:** 10.3390/nu17111881

**Published:** 2025-05-30

**Authors:** Giuseppe Cerullo, Martino V. Franchi, Alessandro Sampieri, Francesco Campa, Antonio Paoli

**Affiliations:** Department of Biomedical Sciences, University of Padua, 35122 Padua, Italy; giuseppe.cerullo@unipd.it (G.C.); martino.franchi@unipd.it (M.V.F.); alessandro.sampieri@phd.unipd.it (A.S.); antonio.paoli@unipd.it (A.P.)

**Keywords:** adipose tissue, anthropometry, body composition, echography, fat mass, four-component model, nutritional assessment

## Abstract

**Background:** The evaluation of body composition is considered a key factor for assessing nutritional status. In several settings, ultrasound (US) has been used as a useful tool in nutritional practice by estimating body composition parameters, such as the whole-body fat mass (FM). The estimation of FM can be carried out by using predictive equations that generally require measurements of skinfold thickness, which can be measured directly via US imaging. The main aim of this study was to evaluate the validity of US-derived skinfolds within anthropometric equations for estimating whole-body FM. **Methods**: Skinfold thickness was measured in 37 active individuals (19 males, age 24.2 ± 4.3 years, and 18 females, age 25.3 ± 4.2 years) using both anthropometry and US. The skinfolds obtained from anthropometry and US were entered into Evans’ equation to estimate the FM and were validated against a four-component model (4C) as a reference. **Results**: The use of US-derived skinfolds within anthropometric equations resulted in an overestimation of FM (4.8%, *p* < 0.01). An agreement analysis between the FMs estimated with US-derived skinfolds and the 4C model revealed a concordance correlation coefficient of 0.33, 95% limits of agreements ranging from −3.4% to 0.6%, and a positive trend (r = 0.8; *p* < 0.01). **Conclusions**: The practice of doubling the US thickness to approximate skinfold thickness leads to an overestimation of FM by ~5%, and it should be avoided. This results in a lack of agreement with the 4C model at both the group and individual levels. New equations based on US measurements should be developed to enhance the accuracy of body composition evaluation and help optimize nutritional strategies.

## 1. Introduction

The assessment of body composition is widely recognized as a fundamental component of the evaluation of nutritional status, providing information that supports the identification, diagnosis, and management of many conditions requiring personalized nutritional interventions [1,2]. The primary aim of assessing body composition is to evaluate nutritional status by quantifying the body fat mass (FM) and fat-free mass, which includes bones, organs, and body water. For instance, in clinical practice, this is especially important for hospitalized patients, as optimizing nutrition helps preserve fat-free mass. Higher fat-free mass indices have been associated with better clinical outcomes, including improved survival and reduced risk of death and hospitalization in patients with heart failure [3]. It is well known that changes in energy balance lead to a reduction or accumulation of body fat mass (FM) in the case of a negative or positive energy balance, respectively [4]. Assessing body fat is a crucial aspect of body composition monitoring, as excessive fat levels can negatively impact health in multiple ways. Excessive adipose tissue is strictly linked to metabolic dysfunctions and reductions in mobility and physical function in general [5,6]. Various techniques are available for quantifying body fat, each operating at different levels of investigation. At the anatomical level, imaging methods such as magnetic resonance imaging (MRI), computed tomography, and ultrasound imaging (US) allow for visualizations of adipose tissue using magnetic fields, radiation, or sound waves, respectively [7]. Complementing these methods, densitometric techniques such as dual-energy X-ray absorptiometry (DXA) are employed to quantify FM (which equates to the lipidic component of the adipose tissue) [8] by analyzing the differential absorption of X-rays across various tissues [7]. Simpler alternatives, such as anthropometric measurements of skinfold thickness, allow for the estimation of body fat by measuring the thickness of a double layer of skin and the related subcutaneous adipose tissue [7]. For this reason, this method is generally criticized in cases of a high level of visceral fat. However, these methods vary in cost, invasiveness, ease of use, and the type of data they provide, necessitating careful selection based on specific assessment requirements.

While there is currently not a universally recognized best method for the assessment of nutritional status, a growing body of evidence suggests that US may also serve as a nutritional assessment and nutritional management tool [9,10,11,12]. According to several authors, the use of US should be encouraged in nutrition practice to assess nutritional risk and to monitor responses to nutritional interventions [10,11,13,14].

Tissue US stands out as a non-invasive, cost-effective method for assessing subcutaneous tissues, with portable devices enabling its use beyond laboratory settings [10]. B-mode US use has increased in the last 30 years for skeletal muscle imaging [15,16]. US could also be particularly valuable for assessing fat distribution between superficial subcutaneous adipose tissue (SSAT) and deep subcutaneous adipose tissue (DSAT). Furthermore, US can isolate adipose tissue by excluding fibrous components, enhancing its accuracy for monitoring fat loss in both clinical and sports contexts [17,18]. US demonstrates a superior predictive role compared to other fat-related measurements, such as skinfolds, exhibiting strong correlations with DXA-derived fat mass [19] and MRI assessments of single anatomical sites [20]. Additionally, US exhibits excellent day-to-day reliability for fat thickness values [21] and remains unaffected by hydration status or food intake [22,23], factors that instead can influence the outcomes obtained by DXA or bioimpedance methodologies [24,25]. For these reasons, US can be seen as a compelling alternative to traditional field methods, with the potential to replace them in both research and more applied settings [10,26].

Beyond qualitative assessments, researchers and practitioners often aim to quantify body fat using predictive equations, which should rely on population-specific models to ensure accuracy [27]. While US-based equations for whole-body FM estimations are gaining interest, their development lacks standardized guidelines, reflecting the diversity of procedures employed. For example, some studies employed regression models based on adipose thickness measured between the lower portion of the dermis and the upper portion of the muscle fascia [19,28]. Other authors focused exclusively on SSAT or DSAT [19], while others considered all tissues from the epidermis to the muscle fascia [29]. This variability complicates efforts to create universally applicable US-based equations. To address these inconsistencies, previous studies attempted to derive anthropometric skinfold thickness from US measurements, aiming to use US-derived skinfolds in anthropometric-based predictive equations. This approach stems from the numerous currently available anthropometric equations, most of which are specifically tailored to different populations, including athletes [27]. Particularly, skinfold thickness refers to the measurement of a double layer of adipose tissue that is interposed between a double layer of skin, obtained by elevating the tissue using the fingers and compressing it with an anthropometric caliper [27,30]. Starting from US measures (Figure 1), skinfolds are commonly calculated by doubling the thickness from the epidermis to the deep fascial membrane [31,32,33,34].

Nevertheless, calculating skinfolds using US measurements has been criticized recently. Particularly, the compressibility characteristics of adipose tissue could potentially lead to inaccuracies [35]. Indeed, when subjected to pressure from fingers and a caliper, the thickness of the compressed tissue appears to be less than twice the thickness under other conditions [20,26]. It is important to emphasize that several factors influence the compressibility of subcutaneous adipose tissue, including individual characteristics and age-related thinning [36,37,38,39]. These factors complicate the establishment of universal calculation coefficients. Despite these challenges, some authors attempted to validate the use of US-derived skinfolds in anthropometric equations to estimate FM [31,32,40]. Unfortunately, they have not employed appropriate reference methods in their study design.

For example, such studies employed DXA or a four-component (4C) model to validate the Jackson and Pollock equation [41], even though this formula was originally developed using a different criterion method (hydrostatic weighing). This represents a methodological limitation, as methods such as DXA, 4C, and hydrostatic weighing exhibit a lack of agreement between them [42,43]. In other words, this methodological mismatch compromises the validity of such comparative studies. Currently, the 4C model is considered the gold standard method for assessing fat mass by dividing the body mass into fat, water, bone mineral content, and residuals, using multiple measurement techniques to minimize errors and account for individual variability [44].

Given the significant potential of US in body composition analysis, determining whether US-derived skinfolds are valid for use in anthropometric equations is crucial for advancing US methodology. Therefore, the present study aimed to validate the use of US-derived skinfolds in anthropometric equations for FM assessment, employing methodologically rigorous procedures to ensure accuracy and reliability. Our hypothesis was that the skinfold thickness does not correspond to a double layer of adipose tissue interposed with other superficial tissues. For this reason, a secondary aim was to determine whether it is possible to identify a new and valid predictive model to estimate skinfold thickness from US measurements.

## 2. Materials and Methods

An appropriate predictive equation was chosen for estimating FM in young adults according to the guidelines proposed in a recent review [27]. Subjects of both genders were included. The following inclusion criteria were used: (i) age of 16 y or older and (ii) a body mass index < 30 kg/m^2^. The therapeutic use of medication, the presence of injuries, or chronic diseases of a metabolic, autoimmune, cardiovascular, or oncological nature were reasons for exclusion from the study. The equation of Evans et al. [45], designed for active populations and mixed athletes, was used. In its simplified form (using 3 skinfolds instead of 7, because two sites—chest and midaxillary—are no longer part of the latest ISAK update [46]), this approach requires measuring the thickness of the triceps, abdominal, and thigh skinfolds. Since Evans’s equation was developed using a 4C model, the same procedure was used as the reference method to quantify FM. Agreement analyses were conducted at both group and individual levels to validate the use of US-derived skinfold measurements for estimating FM. A priori power analysis was conducted to determine the sample size using statistical software (G*Power, version 3.1.9.2; Heinrich Heine University, Stuttgart, Germany). We chose the percentage of FM as the reference parameter. Given the study design, requiring a medium-to-large effect size (ES; f^2^ = 0.3), an α level = 05, a warranted power (1 − β) = 0.8, and *N* = 2 number of predictors, the estimated sample size was 36 subjects. The experimental protocol was granted ethical approval by the University of Padova Review Board (approval code: HEC-DSB/02-2023), and all participants provided written informed consent before participation.

### 2.1. Participants

Thirty-seven collegiate students, including nineteen males (age 24.2 ± 4.3 y) and eighteen females (age 25.3 ± 4.2 y), with an average weekly training time of 7.4 [2.2] h volunteered to participate in the study.

### 2.2. Procedures

During the recruitment phase, participants were fully informed about the aims and potential risks of the study before completing a questionnaire to assess their nutritional status. Dietary intake was evaluated using a three-day food record, which included two weekdays and one weekend day, and was subsequently analyzed using nutritional software [47]. After creating a 3-day food diary, participants were instructed to maintain their caloric intake and, in general, their usual dietary habits.

On the day of the experiment, participants came to the laboratory at 10:00 am, having refrained from alcohol or stimulant beverages and fasted for at least 3 h. Their height and body mass were measured using a mechanical scale with a stadiometer (Seca 711, Seca, Hamburg, Germany). The body mass index was calculated as the total body mass (kilograms) divided by the height (meters) squared. Three skinfold thicknesses (triceps, abdominal, and thigh) were measured by a level 3 anthropometrist following the procedures established by the International Society for Advancement of Kinanthropometry (ISAK) [48]. Skinfold thicknesses were measured to the nearest 0.1 mm using a Harpenden caliper (Baty International Ltd., West Sussex, UK). The technical error of measurement (TEM) score was within the 5%. All US measures were carried out using the Vscan Air CL B-Mode ultrasound system (GE HealthCare, Milan, Italy) with a linear array transducer (3–12 MHz) under the general musculoskeletal setting option. An approximately 3–5 mm layer of ultrasound transmission gel (GIMA, Gessate, Italy) was applied on the site, and the probe was transversally centered over inked site marking and rested on the thick layer, avoiding compression of the underlying skin and subcutaneous adipose tissue depth. Caution was taken to minimize any pressure of the transducer on the skin. US images were taken at each measurement site, and the layer from the superior border of the dermis to the upper part of the fascial layer surrounding the adjacent muscle was measured and recorded to 0.1 mm. All US images were saved and analyzed post image acquisition using the calipers tool from the analysis software included in the Vscan Air device app (Vscan Air v.2 GE HealthCare, Milan, Italy). US measurements were then doubled to calculate skinfolds, as in previous studies [31,32,33,34]. All US measurements were acquired by a skilled operator (FC) and supervised/checked by an operator with long-standing US expertise (MVF). The US operator demonstrated an intraclass correlation coefficient of 0.997 for the test–retest analysis of subcutaneous tissue. Figure 2 illustrates an example of the anthropometric and US measurements taken at the selected sites on a study participant.

The equation proposed by Evans [45] was used to estimate fat mass percentage:Fat mass (%) = 8.997 + (0.24658 × sum of triceps, abdominal, and thigh skinfolds) − (6.343 × sex, where 0 is male and 1 is female) − (1.998 × race, where 0 is white and 1 is black).

Subsequently, 4C-derived FM was estimated using the equation by Wang et al. [49]:Fat mass (%) = ((2.748 × body volume − 0.699 × total body water + 1.129 × bone mineral content − 2.051 × body mass)/body mass) × 100.

A modified 4C model was produced using DXA, bioimpedance, and scale data. In this study, body volume and bone mineral content were estimated from DXA parameters [50] and total body water from bioelectrical impedance [51], as reported below.Body volume (L) = fat mass/0.9007 + lean soft mass/1.064 + bone mineral content/2.982.Total body water (kg) = 0.286 + 0.195 × stature^2^/resistance + 0.385 × body mass + 5.086 × sex, where 1 is male and 0 female.

A whole-body DXA scanner (QDR 4500A, Hologic, Marlborough, MA, USA) that operated with software version V8.26a:3.19 was used. System calibration was conducted as specified by the manufacturer.

Foot-to-hand bioimpedance was performed using a single frequency of 50 kHz for the device (BIA 101 BIVA^®^PRO, Akern Systems, Firenze, Italy). The participants were instructed to remove all objects containing metal and to stay in a supine position during the measurements, isolated from the ground and electrical conductors, with their legs abducted at 45°, shoulders abducted at 30° relative to the body midline, and hands pronated [44]. The precision of the bioelectrical device was assessed before each test session; the test–retest coefficient of variation (CV% = standard deviation/mean × 100%) of resistance and reactance were 0.3% and 0.9%, respectively.

### 2.3. Statistical Analysis

Data were analyzed with Jamovi version 0.9.2.9 (Jamovi project, 2018). The Shapiro–Wilk test was used to check the normal distribution of data. Paired sample *t* tests were employed to compare the mean values obtained from the reference technique and from Evans’s equation using anthropometry and US-derived skinfolds. Agreement between FM percentages estimated with the alternative (i.e., anthropometry and US) and reference (i.e., 4C model) methods was determined using the Bland–Altman analysis, Lin’s concordance correlation coefficient (CCC), and McBride’s strength concordance (almost perfect > 0.99; substantial > 0.95 to 0.99; moderate = 0.90–0.95; and poor < 0.90). Linear regression analysis was performed considering anthropometric-derived skinfolds as dependent variables and the US-derived skinfolds as independent variables, including anatomical sites and gender as covariates. The coefficient of determination (R^2^) was classified as follows: values between 0.75 and 1.00 were considered substantial, values between 0.50 and 0.75 were classed as moderate, and values between 0.25 and 0.50 were considered weak [52]. Statistical significance was determined using a *p*-value < 0.05.

## 3. Results

Table 1 reports the body composition characteristics of the participants. The use of the US-derived skinfolds, obtained by doubling the US measurements in the Evans equation, resulted in an overestimation of the FM percentage compared to the reference method (mean difference = 4.8; t = 7.1; 95% confidence interval: 0.6 to 3.4; *p* < 0.01). In contrast, the FM percentage that was calculated using measured manually skinfolds through anthropometry showed no significant difference from the reference value obtained with the 4C model (mean difference = 0.3; t = 0.6; 95% confidence interval: 0.6 to −1.1; *p* = 0.54), as shown in Figure 3.

The ratios between skinfolds and raw US measurements are also reported in Table 1.

Figure 4 shows the R^2^ values considering the raw measurements derived from US (not doubled) as independent variables and the anthropometric-measured skinfold as dependent variable on the three anatomical sites and stratifying the participants by gender.

When considering the anthropometric-derived skinfold as the dependent variable, the regression analysis revealed that raw US measurements accounted for 52% of the variance (R^2^ = 0.52; standard error of estimation = 4.3 mm). Adding anatomical sites as covariates increased the R^2^ to 0.62 (standard error of estimation = 3.85 mm), and including gender further increased it to 0.63 (standard error of estimation = 3.85 mm).

## 4. Discussion

The aim of this study was to assess the validity of FM estimates obtained from US-derived skinfolds when used in anthropometric predictive equations. As hypothesized, the practice of doubling the raw US measures to obtain US-derived skinfolds, calculated from the upper edge of the epidermis to the top of the muscle fascia, did not produce results that were comparable to those obtained via traditional anthropometry. Instead, this approach resulted in an overestimation of FM of 4.8% with respect to the 4C reference model.

Several previous studies have employed the approach of deriving skinfold thickness using US by doubling the thickness measured from the epidermis to the muscle fascia [31,32,33,34]. Our findings demonstrated a lack of agreement, both at the group and individual levels, when US-derived skinfolds were used in anthropometric equations to quantify FM compared to 4C. Specifically, the CCC revealed poor agreement between the FM obtained via the 4C model and that estimated using the Evans equation [45] when US-derived skinfolds were incorporated. Additionally, a Bland–Altman analysis indicated a mean bias, with a tendency to underestimate low FM values and overestimate higher ones. In contrast, no significant bias or trend was observed when standard anthropometric measures were used in the predictive equation. The anatomical characteristics of subcutaneous tissues and their response to compression during the application of calipers must be considered to better understand these results. The anthropometric skinfold thickness reflects a double layer of adipose tissue interposed between two layers of skin [30]. However, adipocytes can change shape under pressure due to their flexible structure, allowing the lipid droplet within the fat cell to deform and distribute the mechanical forces more evenly [35]. For this reason, the thickness that is measured via US, from the upper edge of the epidermis to the top of the muscle fascia, could not represent the exact half of the skinfold thickness, mostly because of such heterogeneous responses of fat tissue to compression. This concept has been previously discussed. According to several authors, the ratio of skinfold thickness to US-derived thickness might be closer to 1.6 rather than 2.0 [26,35]. Our findings showed that the ratio between tissue thickness measured with US and skinfolds varies from 1.2 to 1.7, while previous studies have identified an average value of approximately 1.6 [26,36]. As we observed, this ratio may depend on the specific anatomical site. We could speculate that this ratio may be very low (between 1.2 and 1.3) when the anatomical site is characterized by minimal subcutaneous fat, making adipocytes more susceptible to mechanical pressure. Conversely, when the tissue is thicker or likely has a greater DSAT component, it becomes less compressible, and this ratio approaches twice the resting measurement, without any lifting or compression being applied. Lastly, as highlighted previously [20], when the adipose tissue thickness is high, the skinfold may not include the DSAT, leading the measurement to closely match the raw US measurement. To visually conceptualize this relationship, Figure 5 displays the subcutaneous tissues in different situations: at rest without any pressure (upper-left), under pressure when the tissue is thin (upper-right), when it is thicker with a balanced amount of superficial and deep subcutaneous adipose tissue (lower-left), and when it is very thick, making it impossible for the skinfold to include both layers (lower-right).

This study confirmed that raw US measurements of subcutaneous adipose tissue cannot be assumed to simply represent half of the skinfold thickness. Furthermore, our findings highlighted how the limited predictive power of such measures complicates the identification of a reliable conversion coefficient for anthropometric skinfold predictions. The unexplained variability in the final regression model (approximately 40%) could stem from factors such as the morphological characteristics of adipose tissue and the presence of connective tissue within the measurement area. Moreover, subcutaneous adipose tissue can be divided into two distinct layers: the SSAT and the DSAT, separated by a connective tissue fascia (Figure 1) [53]. These two layers differ structurally, functionally, and in their relevance to health. Structurally, SSAT has a dense, lamellar arrangement of compact adipocytes that are organized in parallel, resulting in a more structured and less infiltrated tissue [53,54]. Conversely, DSAT exhibits a looser, areolar arrangement with higher infiltration of inflammatory cells. The variable thickness and distribution of connective tissue within SAT can also affect the compressibility, both between individuals and across anatomical sites. In our study, among the three anatomical sites analyzed, the thigh exhibited the strongest relationship between anthropometric and US measurements, suggesting better predictive potential for this site using raw US measures. This may indicate reduced compressibility at the thigh compared to other sites. Although investigating these aspects was not the primary aim of our study, both anthropometric and US-derived skinfold measurements of the thigh were more similar between each other, and this was the site where a relatively thicker DSAT layer appeared to be present (Figure 2). Future studies should explore whether these characteristics are attributable to variations in DSAT thickness, the ratio between SSAT and DSAT, and their responses to external pressure. It is also worth noting that DSAT increases in thickness with higher overall body fat [20], which in some individuals may prevent it from being fully compressed during anthropometric skinfold measurements [20]. This could result in anthropometric skinfolds resembling raw US measures, as demonstrated by MRI-derived imaging [20]. Nevertheless, our findings discourage the development of conversion factors to derive skinfolds from US measurements. Instead, future efforts should focus on creating new predictive equations based on raw US measurements to estimate FM and other body composition parameters.

The strengths of this study include the use of a reference model for FM estimation that is closely aligned with the one used in the development of the equation employed (Evans equation). In fact, the Evans equation [45] was developed against a 4C model, which is currently regarded as the state-of-the-art method for FM quantification [8,55]. In this study, outputs obtained using comparable procedures were compared. Specifically, the equations that were used to estimate FM from skinfolds and US were developed using the 4C model as the criterion method, the same reference that was used in the validation procedures of this study. Based on this, future studies should be cautious when comparing estimates derived from equations that have been developed using different reference methods than those used in their validation (e.g., using formulas developed with DXA and comparing them with 4C-derived values, or vice versa). Moreover, whenever possible, multicomponent models such as the 4C model should be preferred for the development of predictive equations, whether based on anthropometry or ultrasound. Further, a wireless US device was employed to measure the relevant subcutaneous layers, offering a cost-effective and portable alternative to other US technologies. Moreover, approximately 40% of the variability in the relationship between raw US measurements and anthropometric skinfolds remained unexplained. Based on that, further studies using alternative modeling approaches (e.g., machine learning) could improve the predictive accuracy. Lastly, the findings of the present study cannot be generalized to other populations, such as individuals with obesity, who may exhibit different proportions of SSAT and DSAT. Furthermore, future studies should include different populations, ranging from clinical settings to athletes.

### Nutritional Perspectives

The present study provides the first evidence that the use of US-derived skinfolds in anthropometric equations should be discouraged, as it leads to an overestimation of FM. Such overestimations compromise the clinical utility of body composition assessments. For example, this can influence nutritional practice in several ways. First, body composition assessment is widely regarded as the cornerstone for determining nutritional status, providing valuable information that supports both the diagnosis and management of various physiological and pathological conditions where optimized nutrition is essential [1,56]. For instance, the diagnosis of malnutrition, ranging from clinical settings to sports performance, requires the concurrent evaluation of fat-free mass and FM. Both malnutrition and a low body mass index, characterized by reductions in fat-free mass and FM, are associated with increased risks of hospitalization and higher mortality rates, as observed in rehabilitation patients with stable chronic obstructive pulmonary disease [57]. Second, US can assess several parameters that are critical for the evaluation of malnutrition, including a low skeletal muscle mass index [58,59], unintentional body mass loss [60], and decreased food intake [61]. All these elements are key criteria within the Global Leadership Initiative on Malnutrition (GLIM) framework [62]. Furthermore, according to expert panels, bedside techniques such as US may facilitate the individualization of protein intake, even within the intensive care unit setting [14]. Indeed, US shows promise in determining energy intake and assessing protein adequacy [13,63,64,65]. Third, although bioelectrical impedance analysis has been validated as a reliable tool for body composition assessment, its use is limited in patients with pacemakers, and it remains sensitive to fluctuations in body fluids. Alterations in body fluid balance are frequently observed in hospitalized patients, as well as in athletes engaged in high-intensity activities. In particular, track and field athletes experience considerable physiological demands and thus require finely tuned nutritional strategies, especially given the high prevalence of low energy availability that is frequently reported in this population [66]. In this sense, it would be highly desirable to employ an instrument that is capable of accurately assessing FM, independent of hydration status variations.

These results highlight the need for developing new equations based specifically on US measurements, which could enhance the accuracy of body composition evaluation in nutritional assessments in different settings.

## 5. Conclusions

The use of US-derived skinfolds (based on doubling the thickness measured from the upper edge of the epidermis to the top of the muscle fascia) in anthropometric equations leads to an overestimation of fat mass. This practice should therefore be discouraged in research settings, as well as in software developed for practitioners. Rather than seeking to retrofit US data into legacy anthropometric models, the development of novel, US-specific predictive models is warranted. Our findings have direct implications for practitioners in the field, potentially enhancing the accuracy of body composition assessments and helping to tailor nutritional strategies across a range of conditions, from hospitalized patients to elite athletes.

## Figures and Tables

**Figure 1 nutrients-17-01881-f001:**
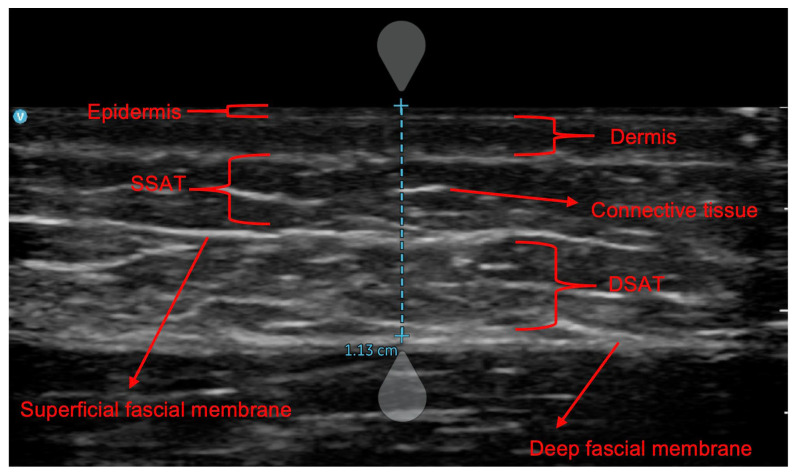
Ultrasound-derived image of subcutaneous tissues at the abdominal site: SSAT (superficial subcutaneous adipose tissue) and DSAT (deep subcutaneous adipose tissue). The blue line identifies the layers that are typically doubled to calculate the skinfold thickness (e.g., skinfold = 11.3 mm × 2 = 22.6 mm, based on the data shown).

**Figure 2 nutrients-17-01881-f002:**
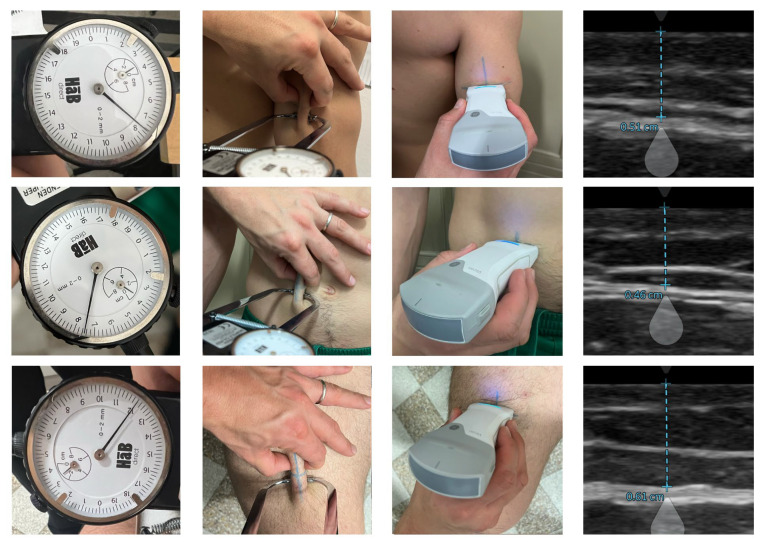
Anthropometric and US-derived measurements.

**Figure 3 nutrients-17-01881-f003:**
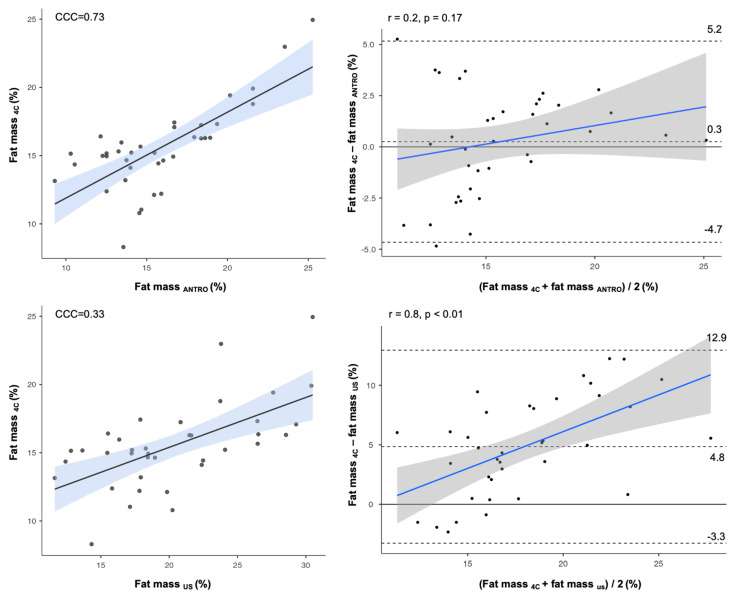
On the left side, scatterplots and concordance correlation coefficients (CCCs) are shown for the fat mass percentages estimated using the Evans equation with anthropometric (ANTRO) measurements and US-derived measurements. On the right side, the results of the Bland–Altman analyses are presented, where the three dashed lines represent, from top to bottom, the upper 95% limit of agreement, the bias, and the lower 95% limit of agreement, respectively. r = coefficient of correlation. Fat mass estimated using 4C represents the reference method.

**Figure 4 nutrients-17-01881-f004:**
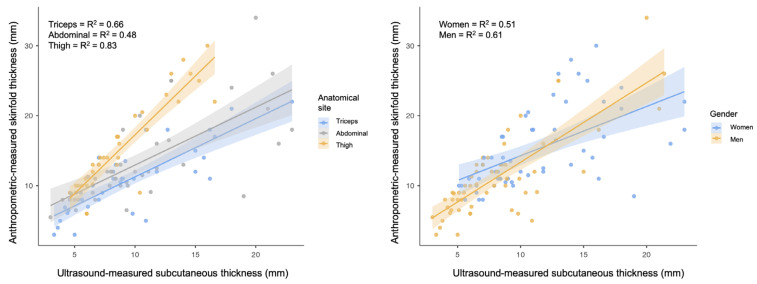
Scatterplots showing the results of the regression analysis for each anatomical site (on the left) and stratifying the participants by gender (on the right). R^2^= coefficient of determination.

**Figure 5 nutrients-17-01881-f005:**
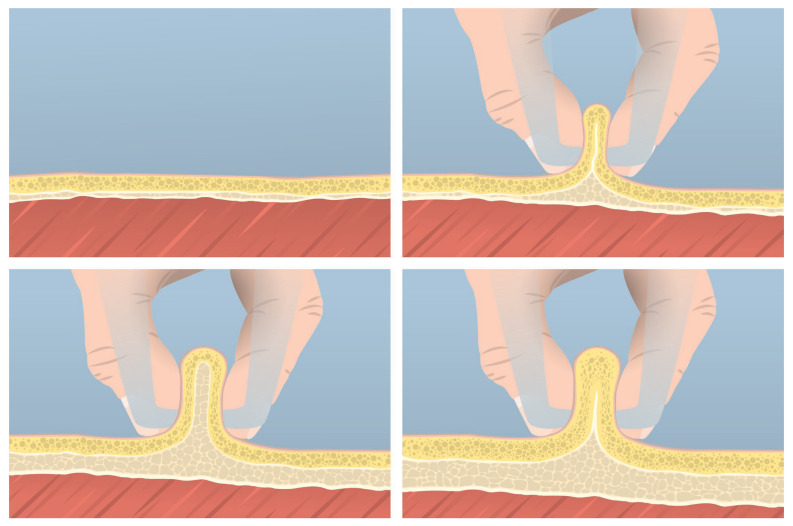
Subcutaneous tissues in different conditions: at rest (**upper-left**), under pressure when the tissue is thin (**upper-right**), in case of a balanced amount of superficial and deep subcutaneous adipose tissue (**lower-left**), and when it is very thick (**lower-right**).

**Table 1 nutrients-17-01881-t001:** Body composition characteristics of the participants.

	Men (*n* = 19) Mean ± SD	Women (*n* = 18) Mean ± SD
Body mass (kg)	79.1 ± 9.7	57.4 ± 5.7
Height (cm)	178.7 ± 6.3	164.9 ± 5.4
Body mass index (kg/m^2^)	24.8 ± 2.7	21.1 ± 1.3
Triceps SKF (mm)	8.4 ± 4.7	13.4 ± 4.2
Abdominal SKF (mm)	14.2 ± 8.0	13.1 ± 4.2
Thigh SKF (mm)	11.3 ± 3.9	19.3 ± 6.3
US-derived triceps SKF (mm)	14.9 ± 7.4	23.4 ± 9.5
US-derived abdominal SKF (mm)	20.5 ± 11.0	23.7 ± 10.5
US-derived thigh SKF (mm)	13.7 ± 3.8	21.6 ± 7.6
Triceps SKF/Raw US measure	1.0 ± 0.4	1.2 ± 0.3
Abdominal SKF/Raw US measure	1.4 ± 0.4	1.2 ± 0.4
Thigh SKF/Raw US measure	1.7 ± 0.4	1.8 ± 0.2
Total body water (l)	50.1 ± 5.3	32.4 ± 2.9
Body volume (l)	75.3 ± 9.8	55.1 ± 5.1
Bone mineral content (kg)	2.8 ± 0.4	2.1 ± 0.4
Lean soft mass (kg)	60.6 ± 6.9	40.7 ± 6.9
Fat mass 4C (%)	15.6 ± 4.1	15.4 ± 1.6
Fat mass DXA (%)	19.4 ± 5.4	24.9 ± 6.0

Note: SD: standard deviation; 4C = four-component model; DXA = dual-energy X-ray absorptiometry; SKF = skinfold.

## Data Availability

The data presented in this study are available on request from the corresponding author. The data is not publicly available due to intellectual property concerns and the proprietary nature of the dataset.

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
