# Peer review of "Ultrasound-Derived Skinfolds in Anthropometric Predictive Equations Overestimate Fat Mass: A Validation Study Using a Four-Component Model"

_nutrients, 2025, doi:10.3390/nu17111881_

Round 1
Reviewer 1 Report
Comments and Suggestions for Authors
The manuscript addresses an essential aspect of nutritional assessment by exploring the validity of ultrasound (US)- derived skinfold measurements in estimating body composition. Given the increasing interest in noninvasive techniques, this research is timely and relevant. Improving methodological details is crucial; clearly defining participant recruitment criteria and identifying potential confounding variables will enhance understanding of your study's context. Another important aspect is to improve the language quality; reviewing the manuscript for grammatical errors and awkward phrasing will significantly enhance readability, and seeking feedback from a colleague or a professional editor can further refine the language. When it comes to suggesting the development of new equations, providing clear future directions is vital; outlining specific methodologies or approaches for future research adds practical value to your conclusions and guides subsequent studies. Finally, it is essential to consider sample size and diversity; acknowledging any limitations and discussing plans for future studies to include a larger and more diverse population will enhance the generalizability of your findings.
Comments on the Quality of English LanguageThe quality of the English language in the manuscript could benefit from several improvements. First, a thorough review of grammatical errors would enhance the clarity of the text. Correcting any awkward phrasing that might confuse readers or detract from the overall message is also essential. Additionally, employing a more varied vocabulary can make the writing more engaging and precise. Consider checking for consistency in terminology and ensuring that technical jargon is clearly defined for readers who may not be specialists in the field.
Seeking feedback from colleagues or professionals also provides valuable insights into unclear sections, further refining the language and improving overall readability. Attention to these details will significantly enhance the manuscript's quality and impact.
Author Response
Comments 1) The manuscript addresses an essential aspect of nutritional assessment by exploring the validity of ultrasound (US)- derived skinfold measurements in estimating body composition. Given the increasing interest in noninvasive techniques, this research is timely and relevant.
Improving methodological details is crucial; clearly defining participant recruitment criteria and identifying potential confounding variables will enhance understanding of your study's context.
Response 1) Thanks for the comment. Inclusion and exclusion criteria have been described in detail. See lines 138-141
Comments 2) Another important aspect is to improve the language quality; reviewing the manuscript for grammatical errors and awkward phrasing will significantly enhance readability, and seeking feedback from a colleague or a professional editor can further refine the language.
Response 2) Thank you for the suggestion. The text has been revised to enhance its overall quality
Comments 3) When it comes to suggesting the development of new equations, providing clear future directions is vital; outlining specific methodologies or approaches for future research adds practical value to your conclusions and guides subsequent studies.
Response 3) We have added further guidance on future directions. See lines 365-373 and 376-378.
Comments 4) Finally, it is essential to consider sample size and diversity;
Response 4) Thanks for the suggestion. Details regarding the sample size are reported in lines 149-153.
Comments 5) Acknowledging any limitations and discussing plans for future studies to include a larger and more diverse population will enhance the generalizability of your findings.
Response 5) More details regarding future studies have been added in lines 381-382.
Comments on the Quality of English Language
The quality of the English language in the manuscript could benefit from several improvements. First, a thorough review of grammatical errors would enhance the clarity of the text. Correcting any awkward phrasing that might confuse readers or detract from the overall message is also essential. Additionally, employing a more varied vocabulary can make the writing more engaging and precise. Consider checking for consistency in terminology and ensuring that technical jargon is clearly defined for readers who may not be specialists in the field. Seeking feedback from colleagues or professionals also provides valuable insights into unclear sections, further refining the language and improving overall readability. Attention to these details will significantly enhance the manuscript's quality and impact.
The text has been revised according your suggestion.
Reviewer 2 Report
Comments and Suggestions for Authors
General Assessment
This manuscript addresses a relevant issue in the field of body composition assessment: the validity of using ultrasound (US)-derived skinfolds within anthropometric predictive equations. The study is methodologically sound, includes a strong rationale, and uses appropriate statistical analyses. However, several issues related to structure, clarity, and interpretation need attention.
Major Comments
1. Title and Abstract
Lines 2–4: The title is descriptive but overly long. Consider simplifying to:
"Validation of Ultrasound-Derived Skinfolds in Predicting Fat Mass: A Comparison with a Four-Component Model"
Lines 9–28: The abstract lacks clarity in a few places. Specifically:
Line 14: “can be obtained through US” → Consider specifying "measured directly via ultrasound imaging".
Lines 24–26: "Duplicate the raw US measurement values" is ambiguous. Clarify that this refers to the practice of doubling US thickness to approximate skinfold thickness.
Line 27: The call for “new equations” is valid but could be strengthened by mentioning that this approach may replace reliance on traditional skinfold-based estimates.
2. Introduction
Lines 33–34: Consider clarifying the scope: “offering critical insights into health risk stratification, diagnosis, and monitoring of nutritional interventions.”
Line 53: Typo: “not a universally recognized best methods” → revise to “not a universally recognized best method”.
Lines 86–91: The explanation of US-derived skinfold calculation is clear, but Figure 1 should explicitly link to this text.
Lines 108–115: Important methodological concern raised about misalignment of validation techniques. Suggest strengthening with a clear summary sentence:
“This methodological mismatch compromises the validity of such comparative studies.”
3. Methods
Lines 126–133: The rationale for choosing the Evans equation is appropriate, but you should justify further why the simplified 3-site version was used over the full 7-site version.
Lines 160–161: ISAK standards are mentioned, but please specify the technician's certification level more explicitly (e.g., ISAK Level 3 certified anthropometrist).
Lines 174–177: Excellent detail on US acquisition. However, include a brief justification for doubling US values—perhaps in a footnote or parenthesis.
Lines 204–217: The derivation of the 4C model is well-explained, but a flowchart or schematic of the full pipeline (US, anthropometry, DXA, BIA → 4C) would improve reader understanding.
4. Results
Line 236: “95% confidence interval: 3.4 to 0.6” → These bounds seem reversed. Should be from 0.6 to 3.4.
Line 241: Figure 3 is helpful. Consider labeling axes more clearly to indicate reference vs. test methods.
Lines 251–252: The ratio of US to skinfold by site is valuable but would benefit from inclusion in Table 1 or its own supplementary table for clarity.
5. Discussion
Lines 274–281: The statement that ~40% of variability remains unexplained is important. Consider discussing whether alternative modeling approaches (e.g., machine learning) could improve predictive accuracy.
Lines 294–299: Strong explanation of biomechanical tissue differences, though the reference to Figure 2 is slightly confusing. Consider moving or renaming the figure for better alignment.
Lines 316–322: Figure 5 is insightful but should be more prominently referred to in the paragraph. It might help to preface this paragraph with a sentence like:
“To visually conceptualize this relationship, Figure 5 illustrates... ”
Lines 351–353: The conclusion here is critical and well-justified. It clearly sets the stage for new equation development using US-only data.
6. Nutritional Implications
Lines 365–396: This section effectively expands the relevance of findings. However:
Line 367: "Undermining their validity" → consider rewording to: "compromising the clinical utility of body composition assessments."
Lines 379–385: The value of US in nutritional settings is valid. Adding a comparison table of field techniques (US vs. BIA vs. DXA) might further strengthen this section.
7. Conclusion
Lines 403–414: Strong conclusion. Lines 408–410 could be more assertive:
Suggest: “Rather than seeking to retro-fit ultrasound data into legacy anthropometric models, the development of novel, US-specific predictive models is warranted.”
Author Response
General Assessment
This manuscript addresses a relevant issue in the field of body composition assessment: the validity of using ultrasound (US)-derived skinfolds within anthropometric predictive equations. The study is methodologically sound, includes a strong rationale, and uses appropriate statistical analyses. However, several issues related to structure, clarity, and interpretation need attention.
Major Comments
- Title and Abstract
Comments 1) Lines 2–4: The title is descriptive but overly long. Consider simplifying to:
"Validation of Ultrasound-Derived Skinfolds in Predicting Fat Mass: A Comparison with a Four-Component Model"
Response 1) Excellent suggestion; the title has been revised.
Comments 2) Lines 9–28: The abstract lacks clarity in a few places. Specifically: Line 14: “can be obtained through US” → Consider specifying "measured directly via ultrasound imaging".
Response 2) Thanks for the comment. Check the new abstract.
Comments 3) Lines 24–26: "Duplicate the raw US measurement values" is ambiguous. Clarify that this refers to the practice of doubling US thickness to approximate skinfold thickness.
Response 3) The sentence has been modified. See lines 25-26
Comments 4) Line 27: The call for “new equations” is valid but could be strengthened by mentioning that this approach may replace reliance on traditional skinfold-based estimates.
Response 4) We revised the text, as suggested.
- Introduction
Comments 5) Lines 33–34: Consider clarifying the scope: “offering critical insights into health risk stratification, diagnosis, and monitoring of nutritional interventions.”
Response 5) Thanks for this comment. We have improved the text by providing an example. See lines 37-42
Comments 6) Line 53: Typo: “not a universally recognized best methods” → revise to “not a universally recognized best method”.
Response 6) Ok, done.
Comments 7) Lines 86–91: The explanation of US-derived skinfold calculation is clear, but Figure 1 should explicitly link to this text.??
Response 7) Thank you for the clarification. We have linked the figure at a different point in the manuscript.
Comments 8) Lines 108–115: Important methodological concern raised about misalignment of validation techniques. Suggest strengthening with a clear summary sentence:
“This methodological mismatch compromises the validity of such comparative studies.”
Response 8) Ok, check lines 120-124.
- Methods
Comments 9) Lines 126–133: The rationale for choosing the Evans equation is appropriate, but you should justify further why the simplified 3-site version was used over the full 7-site version.
Response 9) Thank you for your comment. We chose the 3-skinfold method because the 7-skinfold version includes two sites (chest and midaxillary) that are no longer part of the latest ISAK update (Esparza-Ros F, Vaquero-Cristóbal R, Marfell-Jones M. 2019. International Standards for Anthropometric Assessment. Murcia: International Society for the Advancement of Kinanthropometry). We have specified this in the text. See lines 142-144.
Comments 10) Lines 160–161: ISAK standards are mentioned, but please specify the technician's certification level more explicitly (e.g., ISAK Level 3 certified anthropometrist).
Response 10) As mentioned in line 172, skinfold thicknesses were measured by a level 3 anthropometrist.
Comments 11) Lines 174–177: Excellent detail on US acquisition. However, include a brief justification for doubling US values—perhaps in a footnote or parenthesis.
Response 11) As specified in the manuscript, considering the skinfold as twice the US value is an incorrect practice and should be avoided. See lines 94-96. In fact, our hypothesis was that skinfold thickness does not correspond to a double layer of adipose tissue interposed with other superficial tissues.
Comments 12) Lines 204–217: The derivation of the 4C model is well-explained, but a flowchart or schematic of the full pipeline (US, anthropometry, DXA, BIA → 4C) would improve reader understanding.
Response 12) Thank you for the precise suggestion. We hope that this section is now clearer in the revised version of the manuscript. See lines 202-210
- Results
Comments 13) Line 236: “95% confidence interval: 3.4 to 0.6” → These bounds seem reversed. Should be from 0.6 to 3.4.
Response 13) ok
Comments 14) Line 241: Figure 3 is helpful. Consider labeling axes more clearly to indicate reference vs. test methods.
Response 14) Thanks for this comment. Further explanation regarding the reference method has been provided in line 260.
Comments 15) Lines 251–252: The ratio of US to skinfold by site is valuable but would benefit from inclusion in Table 1 or its own supplementary table for clarity.
Response 15) The ratio between SKF and raw US for each anatomical site has been included in table 1.
- Discussion
Comments 16) Lines 274–281: The statement that ~40% of variability remains unexplained is important. Consider discussing whether alternative modeling approaches (e.g., machine learning) could improve predictive accuracy.
Response 16) Thank you for the suggestion. The final part of the discussion has been expanded, also taking into account the comments from the other reviewers. See lines 365-382
Comments 17) Lines 294–299: Strong explanation of biomechanical tissue differences, though the reference to Figure 2 is slightly confusing. Consider moving or renaming the figure for better alignment.
Response 17) Thank for noticing that. We have decided, to reduce confusion, to refer to Figure 2 at a different point in the text.
Comments 18) Lines 316–322: Figure 5 is insightful but should be more prominently referred to in the paragraph. It might help to preface this paragraph with a sentence like:
“To visually conceptualize this relationship, Figure 5 illustrates...
Response 18) Thank you for the suggestion, we have revised the text according to the instructions — see lines 320-325.
Comments 19) Lines 351–353: The conclusion here is critical and well-justified. It clearly sets the stage for new equation development using US-only data.
Response 19) Thanks
- Nutritional Implications
Comments 20) Lines 365–396: This section effectively expands the relevance of findings. However: Line 367: "Undermining their validity" → consider rewording to: "compromising the clinical utility of body composition assessments."
Response 20) Ok, we modified the sentence according to your comment. See line 387
Comments 21) Lines 379–385: The value of US in nutritional settings is valid. Adding a comparison table of field techniques (US vs. BIA vs. DXA) might further strengthen this section.
Response 21) Thank you for this comment, we completely understand your point of view. The reason we did not directly compare skinfolds, DXA, and ultrasound is that doing so would require using equations developed with DXA in order to obtain results that are only hypothetically aligned with each other. Although it would be very interesting to compare the characteristics of the main available techniques, we believe that this is not the objective of our manuscript.
- Conclusion
Comments 22) Lines 403–414: Strong conclusion. Lines 408–410 could be more assertive:
Suggest: “Rather than seeking to retro-fit ultrasound data into legacy anthropometric models, the development of novel, US-specific predictive models is warranted.”
Response 22) Done.
Reviewer 3 Report
Comments and Suggestions for Authors
In manuscript “The use of ultrasound-derived skinfolds in anthropometric predictive equations results in an overestimation of fat mass: a validation study using a four-components model”, authors studied US method in fat mass estimation in comparison to traditional skin fold measuring in adults subjects. Manuscript can be considered for publication in Nutrients after corrections:
Technical issues:
- English language correction is required;
Merit issues:
- Line 50 It should be clearly stated, that this is a method useless in case of visceral fat;
- Give references in lines:107, 292;
- Line 102: should be “less than a half”;
- Line 109: 4component model may mean something completely different depending of branch of science (economy, sociology, medicine) please give description, what exactly is 4C model in the case of this study;
- Line 127: give brief description of guidelines used in the study;
- What about subjects ethnicity/race as this parameter is also a part of Evans equation?; supplement;
- Lines 130-131: why authors measured only 3 folds? The results would be more precise if authors would measure all 7 folds for each subject;
- In measurement with calliper authors pressed folds but – what was clearly stated in methods section- during US measurement authors measured depth of tissues without and additional pressing; would gentle pressing would mimic pressing folds similar to pressing during measurement with calliper? Expand that issue;
- minor English language correction is required;
Author Response
In manuscript “The use of ultrasound-derived skinfolds in anthropometric predictive equations results in an overestimation of fat mass: a validation study using a four-components model”, authors studied US method in fat mass estimation in comparison to traditional skin fold measuring in adults subjects. Manuscript can be considered for publication in Nutrients after corrections:
Technical issues:
- English language correction is required;
Merit issues:
Comments 1) Line 50 It should be clearly stated, that this is a method useless in case of visceral fat;
Response 1) Thanks for noticed that. See lines 55-58
Comments 2) Give references in lines:107, 292;
Response 2) Done. Regarding “no significant bias or trend was observed when standard anthropometric measures were used in the predictive equation.” It is referred to our results.
Comments 3) Line 102: should be “less than a half”; ??
Response 3) Talking about the thickness of the compressed tissue, it seems to be less than twice compared to the rest condition, as we stated.
Comments 4) Line 109: 4component model may mean something completely different depending of branch of science (economy, sociology, medicine) please give description, what exactly is 4C model in the case of this study;
Response 4) Thanks for the comment. The updated version of the manuscript includes a definition of the 4C model. See lines 120-124
Comments 5) Line 127: give brief description of guidelines used in the study;
Response 5) Thanks for the comment. Further details have been added. See lines 138-140
Comments 6) What about subjects ethnicity/race as this parameter is also a part of Evans equation?;
Response 6) This is a very good point. In this study, we always considered the same coefficient (0) since we have no difference between participants.
Comments 7) Lines 130-131: why authors measured only 3 folds? The results would be more precise if authors would measure all 7 folds for each subject;
Response 7) Thank you for your comment. We chose the 3-skinfold method because the 7-skinfold version includes two sites (chest and midaxillary) that are no longer part of the latest ISAK update (Esparza-Ros F, Vaquero-Cristóbal R, Marfell-Jones M. 2019. International Standards for Anthropometric Assessment. Murcia: International Society for the Advancement of Kinanthropometry). We have specified this in the text. See lines 142-144.
Comments 8) In measurement with calliper authors pressed folds but – what was clearly stated in methods section- during US measurement authors measured depth of tissues without and additional pressing; would gentle pressing would mimic pressing folds similar to pressing during measurement with calliper? Expand that issue;
Response 8) Thank you for your comment. As clearly described in the Methods section, skinfold thicknesses were measured using calipers following the ISAK guidelines, which involve grasping and compressing the skinfold to a standardized extent. In contrast, ultrasound measurements were conducted without additional compression, as the technique is designed to assess the uncompressed thickness of the subcutaneous tissue layers.
The caliper method relies on mechanical compression of a double layer of skin and subcutaneous fat, while ultrasound imaging provides a direct visualization of tissue depth without altering its structure. Attempting to replicate the caliper effect with ultrasound could introduce variability and compromise the accuracy of the measurement. For these reasons, and in line with established protocols, we maintained the standard, uncompressed ultrasound technique.